# Interdisciplinary Cooperation between Pharmacists and Nurses—Experiences and Expectations

**DOI:** 10.3390/ijerph191811713

**Published:** 2022-09-16

**Authors:** Magdalena Waszyk-Nowaczyk, Weronika Guzenda, Paweł Dragun, Laura Olsztyńska, Julia Liwarska, Michał Michalak, Jan Ferlak, Mariola Drozd, Renata Sobiechowska

**Affiliations:** 1Pharmacy Practice Division, Department of Pharmaceutical Technology, Poznan University of Medical Sciences, 6 Grunwaldzka Street, 60-780 Poznan, Poland; 2Student’s Pharmaceutical Care Group, Pharmacy Practice Division, Department of Pharmaceutical Technology, Poznan University of Medical Sciences, 6 Grunwaldzka Street, 60-780 Poznan, Poland; 3Department of Computer Science and Statistics, Poznan University of Medical Sciences, 7 Rokietnicka Street, 60-806 Poznan, Poland; 4Department of Pharmaceutical Technology, Poznan University of Medical Sciences, 6 Grunwaldzka Street, 60-780 Poznan, Poland; 5Department of Humanities and Social Medicine, Medical University of Lublin, 20-093 Lublin, Poland; 6Ludwik Błażek Mulidisciplinary Hospital, 97 Poznanska Street, 88-100 Inowrocław, Poland

**Keywords:** pharmacist, nurse, collaboration, pharmaceutical care, community pharmacy

## Abstract

Background: Getting to know the experience gained so far between professions such as pharmacists and nurses allows for introducing changes aimed at better cooperation, and that can improve the quality of patient care. The aim was to obtain the nurses’ opinions on the ongoing cooperation with pharmacists and to analyze the possibilities of cooperation between these groups. Methods: The survey was conducted from January to March 2021 among 124 nurses in Poland. The link to the electronic questionnaire was sent by e-mails sourced from online social groups for nurses. Before completing the questionnaire, each participant was informed about the anonymous research and the purpose of the data obtained. Results: In total, 80.6% of the respondents confirmed that the pharmacist is a reliable advisor in the field of general information about a drug and 60.9% in the field of clinical information about the drug, and 54.8% of the nurses agreed that a pharmacist should carry out such practices as measuring blood pressure or glucose in a community pharmacy, with 70.1% agreeing that a pharmacist should provide pharmaceutical care in a community pharmacy in the future and the most convinced of this were people with a master’s degree. Of the respondents, 74.1% indicated that pharmacist advice should be fully reimbursed by the National Health Fund or another insurance institution. Conclusions: The study showed that the nursing community appreciates the role of pharmacists and has a positive attitude towards cooperation with this professional group. What is more is that it indicates willingness for interdisciplinary cooperation.

## 1. Introduction

Pharmaceutical care (PhC) is the direct, responsible provision of medication-related care for the purpose of achieving definite outcomes that improve a patient’s quality of life. Hepler’s and Strand’s definition is the most well-known definition for PhC, coming from their article “Opportunities and responsibilities in PhC” from 1990 [1]. Since the implementation of this concept, pharmacists have begun to play a very important role in healthcare systems around the world by providing services and expert assistance to their patients in community pharmacies [2]. This has resulted in raising the rank and prestige of the pharmacy profession [2]. In Poland, as in most countries around the world, community pharmacies are easily accessible, both in urban and rural areas, and people do not need to make an appointment to see a pharmacist [3]. According to the data from 10 January 2022, the number of pharmacies in Poland was 11,911, while the number of pharmacists was 26,162 [4]. These professionals are therefore members of the medical profession with unrestricted access, and community pharmacies are often the first place for patients to go when they have minor ailments or symptoms that herald the onset of illness. Pharmacists are highly trusted by the public. In 2019, 90.4% of Poles declared that they trust pharmacists [5]. It is therefore a professional group that serves patients 24 h a day, all year round, providing expert advice as well as essential life-saving medicines. In addition to dispensing medication, pharmacists provide health advice and information on prevention or treatment of various ailments. Therefore, community pharmacies have been the front door of the Polish healthcare system for many years [6].

Many studies have shown that pharmacist–physician collaboration significantly improves clinical outcomes and optimizes patient care [7,8,9,10,11,12]. For a long time, the focus has been on studying the relationship between these two professional groups, while the collaboration between pharmacists and the nursing community has remained unnoticed [13]. It is worth noting that in recent years in Poland, the qualifications of this professional group have expanded significantly. Since 2016, nurses can write prescriptions for medicines containing certain active substances, and in 2020, nursing advice is included in the list of guaranteed benefits [14,15]. Expanding the role of the nursing community coincided with the implementation of Pharmaceutical Profession Act, which provided an opportunity to implement PhC in Poland. As a result of acquiring new competences, these two professional groups will play an increasingly important role in the Polish healthcare system and have an even greater impact than before on the patient’s pharmacotherapy. The cooperation of nurses and pharmacists is therefore attracting more and more attention. Currently, the clinical benefits of cooperation between these two professional groups have been confirmed in many studies around the world [16,17]. Poland is also starting to depart from the stereotypical traditional roles, which define limited functions of pharmacists and nurses.

Therefore, in this study we decided to obtain the opinion of nurses on the previous cooperation with pharmacists and to analyze the possibilities of partnership between these professional groups in the term of implementation of PhC in a community pharmacy. Due to the increased demand for healthcare services, many countries have faced a progressive number of challenging factors such as limited resources, rising healthcare costs, and high patient expectations [18]. Interdisciplinary collaboration is the basis for any well-functioning healthcare system. Examples from many countries have shown that cooperation between the pharmacist and the nursing community can significantly improve the quality of patient care [13]. In addition, cooperation between medical professionals can contribute to reducing the workload of physicians, which benefits patients the most. Therefore, the authors’ of the work also tried to find the answer to the question: is there any possibility for cooperation between nurses and pharmacists in the field of pharmaceutical care in Poland?

## 2. Materials and Methods

The study was carried out among the nurses. The questionnaires were collected from January to March 2021 as an electronic version. The survey was written by the authors of the publication. The actual study was preceded by a pilot study involving 15 standardized questionnaires. A link to electronic questionnaire was sent by emails acquired from online social groups for nurses. Before completing the questionnaire each of the participants was informed about the anonymous study and the purpose of the data obtained. The answers were entered into electronic form by the respondents and the results were downloaded into an Excel sheet. The questionnaire was sent to 300 respondents, and 124 people took part in the study (93.5% women and 6.5% men). Due to the specificity of the topic, 41.3% of respondents completed the questionnaires. The study used an anonymous questionnaire consisting of five parts. The questionnaire contained questions about general respondents’ information, their previous experience in working with pharmacists, current expectations about cooperation, as well as future expectations as a consequence of the implementation of the PhC program and questions regarding the implementation process of PhC. Open and closed questions were used to obtain answers.

The collected data were securely stored in the Department of Pharmaceutical Technology, Pharmacy Practice Division at Poznan University of Medical Sciences. All subjects gave their informed consent for inclusion. The study was conducted in accordance with the Declaration of Helsinki, and the protocol was approved by the Ethics Committee of Poznan University of Medical Sciences (52/21).

The Statistica PL 12 (StatSoft Polska Sp. z o.o., ul. Kraszewskiego 36, 30-110 Kraków, Poland) package was used to perform the statistical analysis. The chi-square test of independence was used for nominal variables as well as the Fisher exact test for small or zero observed frequencies. In the case of interval variables, where the data did not comply with the normal distribution, and in the case of comparing data from the ordinal scale, the Mann–Whitney U test was used to compare the results between the two groups. For more than two samples, the Kruskal–Wallis test was used with Dunn’s post hoc tests to assess the opinions of people included in the study. When developing the Likert scale, the descriptive statistics parameters were used in the ordinal scale to examine the opinions of the respondents. All statistical analyses were performed at α = 0.05. Analyses were carried out among others in terms of sex, age, education, place of living or workplace of the respondents; however, only those statistically significant were presented in the study.

## 3. Results

An indispensable element of any well-functioning healthcare system is the cooperation of various professional medical groups. Therefore, it was decided to obtain the opinion of the nurses about the experience in the field of cooperation with pharmacists.

Sample characteristics: The most numerous groups were people aged 41–50 (36.3%) and over 50 (30.6%), with slightly fewer people under 30 (24.2%) and aged 31–40 (8.9%). Higher education dominated among the respondents (90.8%), including people with a master’s degree (67.7%) and people with a bachelor’s degree (23.1%). In terms of seniority, the largest group included people working in the profession for 21–30 years (31.4%).

### 3.1. Previous Experience in the Field of Cooperation with a Pharmacist

Of all respondents, 80.6% considered the pharmacist to be a reliable advisor in general information about the drug, and the respondents working outside the hospital confirmed it the most (Figure 1, *p* = 0.044), with 60.9% of the respondents considering a pharmacist a reliable advisor in the field of clinical information about the drug. The respondents cooperating with one pharmacist were the most convinced in this topic (*p* = 0.036). In the opinions of 32.3% respondents, the pharmacist does not routinely consult their patients on the safety and proper use of their medications. Most often they were people with secondary medical education (*p* = 0.017). On the other hand, 38.5% of the respondents confirmed that the pharmacist consulted their patients about advisability of using over-the-counter drugs that should be taken in conjunction with other prescribed drugs. The most convinced of this were people with undergraduate education (*p* = 0.022), and 60.5% of the respondents agreed with the statement that the pharmacist routinely consults their patients on the proper storage of drugs, while 25.0% of the respondents stated that the pharmacist does not routinely consult their patients regarding the advisability of use and the mechanism of action of the prescribed drugs. Of these respondents, 54.8% replied that a pharmacist should carry out such practices as measuring blood pressure or blood glucose level in a community pharmacy, with the most common being men (*p* = 0.049). Table 1 and Table 2 present previous nurses’ experience in the field of cooperation with a pharmacist in general and details.

### 3.2. Current Experience in the Field of Cooperation with a Pharmacist

The authors of the study also decided to check the expectations of the nurses in terms of cooperation with a pharmacist, and 58.8% of respondents used the right granted to the nursing community in 2016 to prescribe drugs, but only 25.3% consulted pharmacists when prescribing drugs for their patients. Among the respondents, 39.5% expected a pharmacist in a community pharmacy to take personal responsibility for solving their patients’ drug problems. The most convinced of this was the respondents aged over 50 (*p* = 0.035). More than 85.0% confirmed that they expect a pharmacist to be a specialist in the field of pharmacology and to educate their patients about the advisability of using and the mechanism of action of the prescribed drugs. People from cities with 50,000 to 100,000 inhabitants were the most convinced of this (*p* = 0.027). More than half of the respondents (58.9%) expected pharmacists in a community pharmacy to monitor their patients’ response to treatment and inform them if the patient has any problems and educate them about the symptoms of the disease (57.2%). People who, apart from working in a hospital and outpatient clinic, had a different place of employment, were the most convinced of this (*p* = 0.034). Table 3 and Table 4 present current nurses’ experience in the field of cooperation with a pharmacist in details and general.

### 3.3. Future Expectations of Working with a Pharmacist

Given the entry into force of the Pharmaceutical Profession Act, and the consequent shift in emphasis towards service provision in Polish mainstream pharmacies, the study set out to find out what the nursing community’s expectations of pharmacists are because of the implementation of the PhC program. Briefly, 70.1% of respondents agreed that a pharmacist should provide PhC in a community pharmacy in the future and the most convinced of this were people with a master’s degree (*p* = 0.032); 31.5% of respondents felt that the pharmacist should not be able to see the patient’s medical records. However, in the opinion of 46.8% of the respondents, the pharmacist should have such a possibility. People up to 40 years of age were most convinced of this (*p* = 0.037), and 76.6% of respondents felt that a pharmacist should be able to provide medication use review in a community pharmacy in the future. People aged 31 to 40 were most convinced of this (*p* = 0.013), and 73.4% of respondents agreed with the statement that a pharmacist should be available by telephone to their patients for questions about their medications, with 75.8% of those surveyed agreeing with the statement that the pharmacist should be rewarded for their work on correct patient therapy. The most decisive were people who did not work in a hospital (*p* = 0.004) and people from cities with more than 500,000 inhabitants (*p* = 0.003). Table 5 and Table 6 present future nurses’ expectations in the field of cooperation with a pharmacist in general and in detail.

### 3.4. Pharmaceutical Care Implementation Process

The authors of the study also decided to find out what is the opinion of the nursing community about the process of implementing PhC in our country, and 64.4% of respondents confirm that the advice provided under PhC should not be paid for by the patient. Moreover, 74.1% of the respondents indicated that pharmacist advice should be fully reimbursed by the National Health Fund or another insurance institution (Figure 2).

## 4. Discussion

Pharmaceutical care is professional patient care that is documented, patient-centered, and based on inter-professional cooperation [19]. The current situation in the Polish health system has shown how important the role of the pharmacist is. The COVID-19 pandemic has put tremendous pressure on healthcare systems around the world. Difficult access to physicians has shown that all medical professions, including pharmacists, are of fundamental importance for the functioning of the healthcare system. Due to the widespread availability of pharmacies, they are often the first point of contact between a patient and a physician. In the face of the pandemic, in many countries, not only in Poland, the need for additional pharmaceutical services was noticed, which required changes in legal regulations [20]. Due to the high competence and availability of pharmacists, it is possible to relieve other medical groups and also improve the pharmacotherapy of patients and reduce the phenomenon of polypharmacy. The study showed that only 25.3% nurses who prescribe drugs consult their decisions with pharmacists. Only one-fourth of the respondents consulted the prescribed drugs with pharmacists, probably for a very mundane reason—they did not know that they could expect it from pharmacists. Some nurses may also not feel the need to consult someone on their prescriptions, not considering it important, or have limited access to pharmacists, e.g., in a hospital. This may be due to the fact that in Poland the knowledge among nurses about the role of pharmacists is limited. The reason for this may be the lack of knowledge of other healthcare professionals in the field of pharmacist competences and the possibility of his participation in drug reviews, pharmacological counseling or screening tests. This is confirmed by the studies conducted by Piecuch et al. [21]. They showed that Polish pharmacists cooperate with other medics to a limited extent. On the other hand, in a study conducted in Ontario, 84.0% of doctors consulted with pharmacists up to five times a week, 78.0% sought help from pharmacists regarding the pharmacotherapy of their patients, 28.0% referred their patients for drug reviews, and 44.0% did not know about such services [22]. Some research clearly shows that the cooperation of various medical professions with pharmacists brings significant benefits to the patient during pharmacotherapy [23].

A pharmacist, together with nurses, can perfectly complement a physician, which translates into better patient outcomes [24]. Consultations of a nurse or nurse and pharmacist in primary healthcare resulted in an increase in the correct use of drugs by patients [17]. In an Australian study, establishing cooperation between employees of a community pharmacy and people with nursing education proved effective in caring for patients with mental illnesses. In this innovative project, they were responsible for patient medical records, management, and referrals, while the role of the pharmacist was to manage the drugs. It turned out that the cooperation of pharmacists with the nursing community based on good communication is effective and profitable as it allows to agree on many discrepancies in treatment and thus reduces the occurrence of potential adverse effects and increases the safety of taking medications [23]. Moreover, it was found that joint intervention reduced the number of treatment errors and improved the accuracy of the medications taken [25]. It has been proven also that such cooperation within primary healthcare can improve pain management, reduce resource use and achieve a high level of satisfaction [26]. Schellack et al. reported that integrating nursing educators and pharmacists into antibiotic management programs is effective in improving treatment outcomes and reducing antimicrobial resistance [27].

Most of the respondents, assessing their previous experience stated that pharmacist is a reliable advisor in the field of general information about the drug and, according to 60.9% of respondents, is a righteous aide in the field of clinical information about the drug. In the opinions of the one-third of respondents with secondary medical education, the pharmacist does not routinely consult their patients on safety and proper use of their medications. This result shows that the knowledge of nurses with secondary education about the role of pharmacists is very limited. They consider the pharmacist as someone who dispenses medications to patients. This means that the role of the pharmacist has not been fully understood by healthcare teams yet. Similar results were obtained in a study by Abdul Nabeel Khan et al. [13]. Results showed that the role of pharmacists is underestimated among nurses. To gain favor from other medical professions, pharmacists need to build more positive image and focus PhC on the patient. The respondents confirmed that a pharmacist should carry out such practices as measuring blood pressure and blood glucose in a community pharmacy. This is also confirmed by studies conducted in Canada and the USA. It has been proven that nurses can also intervene in cooperation with pharmacists to improve blood pressure [28].

The study also examined the expectations of the nursing community in terms of collaboration with the pharmacist. Most people confirmed that they would like to have in a pharmacy a specialist in pharmacology. In addition, the majority declared that they would like to educate their patients in community pharmacy about the purpose of use and mechanism of action of the prescribed drugs as well as the course of the disease and to monitor the response to treatment of their patients or to inform if the patient has any problems. This shows that the nursing community is optimistic in its expectations of pharmacists. This result is also supported by a study conducted at the US Trauma Center, in which 76.0% of nurses accepted that prescription orders must be checked by a pharmacist before being processed [29]. Several studies have shown that the above collaboration can minimize many adverse drug reactions and improve the safety of pharmacotherapy [30].

The study also checked the expectations regarding cooperation with a pharmacist as a consequence of the implementation of the PhC. About 70.0% of the respondents agreed with the statement that a pharmacist should run PhC in a community pharmacy, which shows that the nursing community has a positive attitude to cooperation with pharmacists. This result is in line with expectations. Examples of other countries show that this will not solve the medical staff shortage, but will reduce the burden on other professions, which mostly benefits the patient [31]. Moreover, 46.8% of respondents believed the pharmacist should be able to access patient medical records in the future, and 76.6% of the respondents believed that the pharmacist should in the future perform drug checkups in a community pharmacy. These results are in line with the Hughes and Lapane study in which the nursing community was open to accepting pharmacist interventions in nursing homes [32].

One of the elements of practicing PhC is cooperation with representatives of other medical professions. Medical collaboration is the foundation of any well-functioning healthcare system. For several decades, the trend of the model based on interdisciplinary cooperation has been observed. In practice, cooperation between various representatives of medical professions is not easy and requires regulations that clearly define these areas, respecting the autonomy of individual professional groups. For effective partnership, it is necessary to create a model that determines where the responsibility of a given professional group begins and ends [33]. The Polish pharmacy market includes many stakeholders whose goals do not coincide, which makes the implementation of PhC much more difficult [7]. A critical challenge is to develop optimal reimbursement for pharmacy services, shift incentive structures from quantity of service to high quality and standardize services at the state level [13].

## 5. Limitation of the Study

This study included a relatively small sample of respondents, where higher education dominated among subjects. It may result from the adopted research methodology, where the questionnaires were sent by e-mail, but the authors of the study found this to be the best form of contact in times of a pandemic. Moreover, not all nurses answered every question in the survey. It is possible that some of the questions concerned issues that some people have not come across so far. It could also be due to a reluctance to devote more time to a given questionnaire. However, these results showed some trends and directions in the field of pharmaceutical care implementation in Poland. It is worth considering in the future complementary qualitative research which will surely enrich the results of the study

## 6. Conclusions

Interdisciplinary collaboration between health professionals can significantly improve patient care. Over the past few years, pharmacists have begun to establish collaborative relationships with other healthcare professionals [1]. The survey highlighted that the nursing community has a positive perception of the pharmacist’s role in the Polish healthcare system. Nurses have a good experience of working with the pharmacist and consider them to be a valuable source of drug information. This should encourage pharmacists to redefine their role in community pharmacies and show more initiative in order to deepen their collaboration with other staff in the Polish healthcare system, thereby improving the situation of the patient and the state budget. Another aspect is the need to increase pharmacies’ access to technology, mobile health apps and telemedicine tools [14,15]. The COVID-19 pandemic further multiplied the number of benefits provided remotely. This presents an opportunity for pharmacies to move some services to the Internet [13,16]. Interdisciplinary cooperation between representatives of different medical professions is the foundation of any well-functioning healthcare system. Examples from other countries show that cooperation between pharmacists and the nursing community can relieve the burden on the medical community, improve the safety of patient treatment and enhance the quality of medical services. Moreover, such cooperation between medical professionals could positively influence the satisfaction of pharmacists and nurses in everyday practice.

## Figures and Tables

**Figure 1 ijerph-19-11713-f001:**
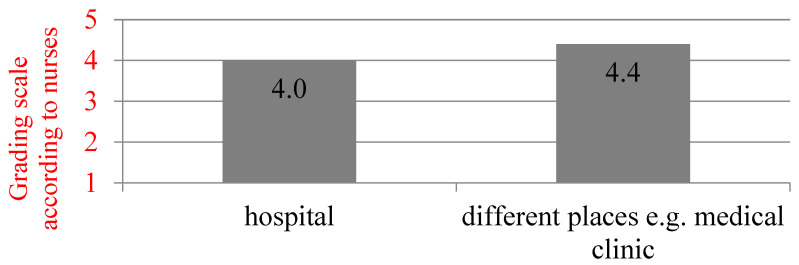
Assessment of the nurses’ opinion of whether the pharmacist is a reliable advisor in the field of general information about the drug, depending on the place of employment (*n* = 124, *p* = 0.044 *, Dunn’s pairwise comparison) Note: * Results statistically significant at α = 0.05.

**Figure 2 ijerph-19-11713-f002:**
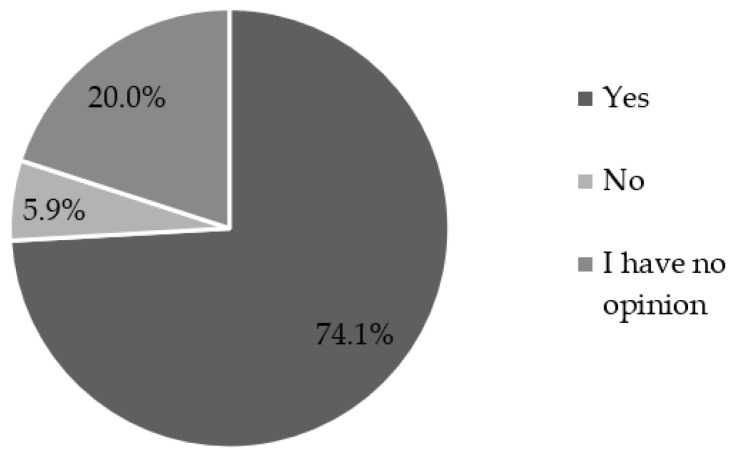
Nurses’ opinion on whether individual documented pharmacist’s advice should be fully reimbursed by the National Health Fund or other insurance institution (*n* = 85).

**Table 1 ijerph-19-11713-t001:** Nurses’ previous experience in the field of cooperation with a pharmacist in general.

Opinion of a Responder Whether:	Strongly Agree	Agree	Neither Agree nor Disagree	Disagree	Strongly Disagree
a pharmacist is a reliable advisor for clinical information about the drug (*n* = 87)	26.4%	34.5%	23.0%	12.6%	3.4%
a pharmacist routinely consults patients on the safety and proper use of medications (*n* = 65)	10.8%	21.5%	35.4%	13.8%	18.5%
a pharmacist routinely consults patients on the advisability of using other over-the-counter medications that should be taken in conjunction with prescribed medications (*n* = 65)	7.7%	30.8%	33.8%	13.8%	13.8%
a pharmacist should conduct pharmacy practices such as measuring blood pressure or blood glucose level (*n* = 124)	24.2%	30.6%	19.4%	13.7%	12.1%
a pharmacist is a reliable advisor for general information about the drug (*n* = 124)	38.7%	41.9%	8.1%	6.5%	4.8%
a pharmacist routinely consults patients on the proper storage of drugs (*n* = 124)	25.8%	34.7%	18.5%	11.3%	9.7%
a pharmacist routinely consults patients on the advisability of using and the mechanism of action of the prescribed drugs (*n* = 124)	14.5%	29.0%	31.5%	15.3%	9.7%

**Table 2 ijerph-19-11713-t002:** Nurses’ previous experience in the field of cooperation with a pharmacist in detail.

Characteristics of Participants *n* (%)						
Grading Scale	1 Strongly Disagree	2 Disagree	3 Neither Agree nor Disagree	4 Agree	5 Strongly Agree	Average
a pharmacist is a reliable advisor in the field of clinical information about the drug (*n* = 87)						
Number of pharmacists working with a nurse						
0	2 (3.1)	10 (15.4)	18 (27.7)	21 (32.3)	14 (21.5)	4.3
1	0 (0.0)	0 (0.0)	0 (0.0)	4 (40.0)	6 (60.0)	4.5
2	1 (20.0)	0 (0.0)	1 (20.0)	2 (40.0)	1 (20.0)	3.4
3	0 (0.0)	0 (0.0)	0 (0.0)	2 (66.7)	1 (33.3)	4.3
>3	0 (0.0)	1 (25.0)	1 (25.0)	1 (25.0)	1 (25.0)	3.5
	*p* = 0.036, Kruskal–Wallis equality-of-populations rank test, Dunn’s pairwise comparison					
a pharmacist routinely consults patients on the safety and proper use of medications (*n* = 65)						
Education						
Higher—master’s degree	8 (18.2)	7 (15.9)	14 (31.8)	10 (22.7)	5 (11.4)	2.9
Higher—bachelor’s degree	0 (0.0)	1 (6.7)	8 (53.3)	3 (20.0)	3 (20.0)	3.5
Medical secondary	4 (66.6)	1 (16.7)	1 (16.7)	0 (0.0)	0 (0.0)	1.5
	*p* = 0.017, Kruskal–Wallis equality-of-populations rank test, Dunn’s pairwise comparison					
a pharmacist routinely consults patients on the advisability of using other over-the-counter drugs that should be taken in conjunction with prescribed drugs (*n* = 65)						
Education						
Higher—master’s degree	5 (11.4)	7 (15.9)	12 (27.3)	17 (38.6)	3 (6.8)	3.1
Higher—bachelor’s degree	0 (0.0)	1 (6.7)	9 (60.0)	2 (13.3)	3 (15.0)	3.5
Medical secondary	4 (66.6)	1 (16.7)	0 (0.0)	1 (16.7)	0 (0.0)	1.7
	*p* = 0.022, Kruskal–Wallis equality-of-populations rank test, Dunn’s pairwise comparison					
a pharmacist should conduct pharmacy practices such as blood pressure or blood glucose measurement (*n* = 124)						
Gender						
Women	15 (13.0)	15 (13.0)	24 (20.9)	35 (30.4)	26 (22.6)	3.4
Men	0	1 (12.5)	0	3 (37.5)	4 (50.0)	4.3
	*p* = 0.049, Two-sample Wilcoxon rank-sum (Mann–Whitney) test					

**Table 3 ijerph-19-11713-t003:** Nurses’ current experience in the field of cooperation with a pharmacist in general (*n* = 124).

Opinion of a Responder Whether Pharmacist in a Community Pharmacy Should:	Strongly Agree	Agree	Neither Agree nor Disagree	Disagree	Strongly Disagree
take personal responsibility for solving his/her patients’ drug problems	21.8%	17.7%	16.9%	32.3%	11.3%
a be a specialist in the field of pharmacology	58.9%	26.6%	11.3%	2.4%	0.8%
educate patients on the advisability of using and the mechanism of action of the recommended drugs	50.8%	37.1%	6.5%	4.8%	0.8%
monitor the response to treatment of patients’ and inform of any problems	31.5%	27.4%	27.4%	12.1%	1.6%
educate patients about the symptoms and course of the patient’s disease	30.6%	26.6%	19.4%	20.2%	3.2%

**Table 4 ijerph-19-11713-t004:** Nurses’ current experience in the field of cooperation with a pharmacist in detail.

Characteristics of Participants *n* (%)						
Grading Scale	1 Strongly Disagree	2 Disagree	3 Neither Agree nor Disagree	4 Agree	5 Strongly Agree	Average
pharmacist takes personal responsibility for solving patients’ drug problems (*n* = 124)						
age [years]						
≤30	1 (3.3)	10 (33.3)	5 (16.7)	9 (30.0)	5 (16.7)	3.2
31–40	3 (27.3)	5 (45.5)	0 (0.0)	2 (18.2)	1 (9.0)	2.4
41–50	8 (18.2)	15 (34.0)	8 (18.2)	5 (11.4)	8 (18.2)	2.8
>50	3 (6.1)	13 (26.5)	12 (24.5)	5 (10.2)	16 (32.7)	3.5
*p* = 0.035, Kruskal–Wallis equality-of-populations rank test, Dunn’s pairwise comparison						
pharmacist educates patients on the advisability of using and the mechanism of action of the recommended drugs (*n* = 124)						
City size [inhabitants]						
≤50,000	1 (4.0)	4 (16.0)	1 (4.0)	9 (36.0)	10 (40.0)	3.9
50,000–100,000	0 (0.0)	0 (0.0)	2 (5.7)	9 (25.7)	24 (68.6)	4.6
>100,000–500,000	0 (0.0)	2 (5.7)	3 (8.6)	16 (45.7)	14 (40.0)	4.2
>500,000	0 (0.0)	0 (0.0)	2 (7.1)	11 (39.3)	15 (53.6)	4.5
*p* = 0.027, Kruskal–Wallis equality-of-populations rank test, Dunn’s pairwise comparison						
pharmacist educates the patients about the symptoms of the disease the patient suffers from (*n* = 124)						
Workplace						
Hospital	4 (3.5)	25 (21.9)	23 (20.2)	29 (25.5)	33 (28.9)	3.5
Other place	0 (0.0)	0 (0.0)	1 (11.1)	3 (33.3)	5 (55.6)	4.4
*p* = 0.034, Kruskal–Wallis equality-of-populations rank test, Dunn’s pairwise comparison						

**Table 5 ijerph-19-11713-t005:** Nurses’ future expectations in the field of cooperation with a pharmacist in general. (*n* = 124).

Opinion of a Responder Whether Pharmacist’s Should in the Future:	Strongly Agree	Agree	Neither Agree nor Disagree	Disagree	Strongly Disagree
provide pharmaceutical care in a community pharmacy	39.5%	30.6%	24.2%	2.4%	3.3%
have access to patient’s medical records	21.0%	25.8%	21.8%	21.0%	10.5%
be allowed to perform drug reviews in a community pharmacy	39.5%	37.1%	17.7%	3.2%	2.4%
be available by telephone to patients when they have questions about medications	32.3%	41.1%	12.1%	9.7%	4.8%
be remunerated for work on patient’s correct therapy	50.8%	25.0%	18.5%	3.2%	2.4%

**Table 6 ijerph-19-11713-t006:** Nurses’ future expectations in the field of cooperation with a pharmacist in detail.

Characteristics of Participants *n* (%)						
Grading Scale	1 Strongly Disagree	2 Disagree	3 Neither Agree nor Disagree	4 Agree	5 Strongly Agree	Average
a pharmacist should provide pharmaceutical care in a community pharmacy (*n* = 65)						
Education						
Higher–master	0 (0.0)	0 (0.0)	9 (20.4)	16 (36.4)	19 (43.2)	4.2
Higher–bachelor’s degree	0 (0.0)	0 (0.0)	5 (33.3)	6 (40.0)	4 (26.7)	3.9
Medical secondary	2 (33.2)	1 (16.7)	1 (16.7)	1 (16.7)	1 (16.7)	2.7
*p* = 0.032, Kruskal-Wallis equality-of-populations rank test, Dunn’s Pairwise Comparison						
a pharmacist should have access to the patient’s medical records (*n* = 124)						
age [years]						
≤30	1 (3.3)	5 (16.7)	6 (20.0)	10 (33.3)	8 (26.7)	3.6
31–40	1 (8.3)	2 (16.7)	3 (25.0)	3 (25.0)	3 (25.0)	3.6
41–50	5 (11.4)	16 (36.4)	9 (20.4)	10 (22.7)	4 (9.1)	2.8
>50	7 (18.4)	3 (7.9)	9 (23.7)	8 (21.1)	11 (28.9)	3.3
*p* = 0.037, Kruskal-Wallis equality-of-populations rank test, Dunn’s Pairwise Comparison						
a pharmacist should be able to perform drug checkups in a community pharmacy (*n* = 124)						
age [years]						
≤30	0 (0.0)	1 (3.3)	3 (10.0)	15 (50.0)	11 (36.7)	4.2
31–40	0 (0.0)	0 (0.0)	1 (9.1)	1 (9.1)	9 (81.8)	4.7
41–50	1 (2.3)	2 (4.5)	14 (31.8)	14 (31.8)	13 (29.6)	3.8
>50	2 (5.3)	1 (2.6)	4 (10.5)	15 (39.5)	16 (42.1)	4.1
*p* = 0.013, Kruskal-Wallis equality-of-populations rank test, Dunn’s Pairwise Comparison						
a pharmacist should be remunerated for working on the correct therapy of the patient (*n* = 124)						
Workplace						
Hospital	3 (3.0)	3 (3.0)	19 (19.2)	27 (27.3)	47 (47.5)	4.1
Other place	0 (0.0)	0 (0.0)	0 (0.0)	0 (0.0)	8 (100.0)	5.0
*p* = 0.004, Kruskal-Wallis equality-of-populations rank test, Dunn’s Pairwise Comparison						

## Data Availability

The data presented in this study are available on request from the corresponding author.

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
