# Peer review of "Interdisciplinary Cooperation between Pharmacists and Nurses—Experiences and Expectations"

_ijerph, 2022, doi:10.3390/ijerph191811713_

Round 1
Reviewer 1 Report
The manuscript entitled “Interdisciplinary cooperation between pharmacists and nurses - experiences and expectations” deals with the collaboration between pharmacists and nurses in Poland. In the study, the authors sought the opinion of nurses about their previous cooperation with pharmacists and analysed the possibilities of partnership between these professions in the implementation of pharmaceutical care in a community pharmacy. The survey is well designed, and the statistical data processing is adequate. The conclusions and limitations of the study are adequately addressed.
The manuscript is recommended for publication in the International Journal of Environmental Research and Public Health after minor revision.
Please consider the following suggestions and corrections:
Line 5: …Ferlak 4, Mariola… With a comma that is not in the superscript.
Line 41: Put the abbreviation in parentheses after the spelled out form the first time it appears in the text: Pharmaceutical care (PhC) is the direct, …
Line 43: Strand’s not Strand's
Line 44: Use quotation marks to enclose the titles of uniquely named parts and sections of a book or a paper: …from their article “Opportunities and responsibilities in pharmaceutical care”, from 1990 [1].
Line 52: was 26,162 instead of - 26,162
Line 52: professionals instead of specialists
Line 125 and elsewhere in the manuscript: Leave a space before and after mathematical operators that function as verbs or conjunctions, i.e., have numbers on both sides or a symbol for a variable on one side and a number on the other. E.g., p < 0.05; n = 124
Line 172, Table 4. and elsewhere in the manuscript: When mathematical symbols are used as adjectives, i.e., with one number that is not part of a mathematical operation, do not leave a space between the symbol and the number. E.g., ≤30; >50
Table 4.: Use an en dash to indicate a range, do not leave a space between the en dash and the number. E.g., 31–40.
Author Response
Thank you very much for your valuable comments and suggestions. All changes have been made in red in the text as recommended.

Reviewer 2 Report
This study addresses an important topic of interdisciplinary cooperation in health care systems to improve patient care and health outcomes. More specifically, the aim of this study was to evaluate the professional relationship between nurses and pharmacists, indeed many times overlooked in other studies and clinical practice. The article is well written, with meaningful introduction and discussion, nevertheless I have some comments, mainly regarding the content, analysis and presentation of the results.
I first report some general feedback and later more specific details:
11. Based on the results it seems this study is more about nurses opinion and beliefs regarding professional cooperation with pharmacists rather than experiences of nurses that have been working together with pharmacists? Perhaps that should be more clear throughout the article. If the questionnaire of this study did include some other results more related to direct experiences that are not yet included in the results, such data might be interesting to add (there are some additional data reported in the discussion that are not in the results?).
22. The statistical analysis performed is univariate statistics, but it is unclear how covariates were selected. The selection of each covariate for each outcome/question should be justified. I suggest carefully reconsidering and justifying statistical analysis and related presentation and discussion.
33. I would suggest putting sample characteristics at the beginning of the results (and excluded from the methods section). I would also expect some discussion regarding sample characteristics (for example is female to male ratio in the sample expected and reflects general population of nurses ect.) I would also suggest some discussion of missing data – some questions have 124 answers and some only 64 – is there some possible explanation?
Introduction
Line 52: »…while the number of pharmacists - 26,162 [4].«
I suggest using »was« instead of - before the number.
Materials and Methods
Line 94: »The actual study was preceded by a pilot study involving 15 forms.«
It is not clear, what kind of forms? Standardized questionnaires perhaps? Or some other forms?
Line 99: »124 people took part in the study (93.5% women and 6.5% men). The questionnaire was sent to 300 respondents. Unfortunately, due to the specificity of the topic, only 41.3% of respondents completed the questionnaires.«
I suggest different order of the information presented – first 300 questionnaires and than 124 responses… In my experiences also 40% response rate is quite ok.
Line 101: »The most numerous groups were people aged 41-50 (36.3%) and over 50 (30.6%), slightly less people under 30 (24.2%) and aged 31- 102 40 (8.9%). Higher education dominated among the respondents (90.8%), including people with a master's degree (67.7%) and people with a bachelor's degree (23.1%). In terms of seniority, the largest group included people working in the profession for 21-30 years (31.4%)«
I suggest putting sample characteristics in the results – first paragraph/chapter.
Line 103/104: Is it expected, that 67% of nurses would have a masters degree? Perhaps some comment is necessary?
Line 122: »For more comparisons, the Kruskal-Wallis test was used with Dunn's post-hoc tests to assess the opinions of people included in the study, an estimated scale was used - the Likert scale.«
Instead of “more comparisons”, perhaps “For more than two samples…”
The last part of the sentence is not clear.
Line 125: “All statistical analyses were performed at p<0.05.”
The correct statement is »All statistical analyses were performed at α=0.05.« Or »All statistical analyses were performed at a significance level 0.05.« Should also be corrected at Figure 1 caption and anywhere else in the article.
Results
Line 133: »p=0.0438«
I believe reporting p values with 3 decimals is usually a standard form of reporting a p value. In this case p=0.044.
Line 133 and others: Beginning of the sentence with a number is not advised. Perhaps allowed in the results but I suggest rephrasing in the discussion.
Line 135: »Of all respondents, 80.6% considered the pharmacist to be a reliable advisor in general information about the drug and the respondents working outside the hospital were the most convinced (Fig.1, p=0.0438).«
Perhaps instead of »more convinced« some other expression should be used e.g. more strongly agreed… or something about belief/opinion?
Line 135: »32.3% of the respondents answered that the pharmacist does not routinely consult their patients on the safety and proper use of their medications«
If I understand correctly the question is more about the respondent’s belief or opinion (agree, disagree), rather than what pharmacists really do? I suggest making that more clear when reporting the results.
Line 150: Figure 1: The y scale should include all possible values (from 1 to 5?), cutting the bottom of the scale visually changes the data presented. It should also be more clearly stated what the y axis presents. The test used should be stated in Figure caption.
Line 154: Table 1: “Previous nurses' experience…”
Is that the correct expression or should it be “nurses' previous experience…”? Also most questions are more about opinion/beliefs than experience – or not?
Line 154: Table 1:
There are some missing data with only some questions? Perhaps that needs some comment in the discussion part.
“A pharmacist routinely consults patients on the safety and correct use of medications (n=65).”
Was that a question based only on the opinion of the responder? A responder saying disagree would mean only his opinion and not necessarily the actual practice? Perhaps that should be emphasised in the table – Perhaps first row, first column should say: “Opinion of a responder/nurse whether:”
Line 155: Table 2: It is not clear what “number of pharmacists” mean – a number of pharmacists a nurse is working with?
How were covariate variables for the statistical analysis selected? For example why education is used in one question and gender in another? The choice of covariate should be justified in the methods section.
Table 1 and 2 have different order of answers – I suggest the same order in all tables.
Line 171: Table 3. Again the questions are about opinion and not experience? I might be wrong of course, perhaps that should be more clearly stated somewhere.
Discussion
Line 216: “The study showed that only 25.3% nurses who prescribe drugs consult their decisions with pharmacists.”
Very interesting and important finding. I am not sure whether this information is from some other study (source?) or this study – in this case I couldn’t find that in the results section?
Line 250-264: This paragraph in the discussion suggests, that the answers in the first two parts of the results are more about opinion/beliefs than experience of nurses with pharmacists.
Line 263: »This is a correct idea«
Perhaps a different wording would be more appropriate.
Line 301: Limitations: I suggest some comment on missing data.
Conclusions
Line 324: »Moreover, such cooperation of medical professionals could positively influence on the satisfaction from everyday practice.«
The sentence is not clear. The satisfactions of patients? Or patients and professionals? (Also without »on«.)
Author Response
Thank you very much for your valuable comments and suggestions. All changes have been made in red in the text as recommended. Please see the attachment.

Round 2
Reviewer 2 Report
Thank you for the revised version of the manuscript.